# Contribution of Adenosine in the Physiological Changes and Injuries Secondary to Exposure to Extreme Oxygen Pressure in Healthy Subjects

**DOI:** 10.3390/biomedicines10092059

**Published:** 2022-08-24

**Authors:** Alain Boussuges, Jeremy Bourenne, Farid Eloufir, Julien Fromonot, Giovanna Mottola, Jean Jacques Risso, Nicolas Vallee, Fabienne Bregeon, Régis Guieu

**Affiliations:** 1Centre for Nutrition and Cardiovascular Disease (C2VN), INSERM, INRAE, Aix Marseille University, 13005 Marseille, France; 2Service d’Explorations Fonctionnelles Respiratoires, CHU Nord, Assistance Publique des Hôpitaux de Marseille, 13005 Marseille, France; 3Department of Intensive Care, Timone University Hospital, 13015 Marseille, France; 4Laboratory of Biochemistry, Timone University Hospital, 13005 Marseille, France; 5Institut de Recherche Biomédicale des Armées, IRBA, Equipe de Recherche Subaquatique Opérationnelle (ERSO), Toulon CEDEX 9, 91220 Brétigny-sur-Orge, France

**Keywords:** Adenosine, hypoxia, altitude, hyperoxia, diving

## Abstract

Climbers and aviators are exposed to severe hypoxia at high altitudes, whereas divers are exposed to hyperoxia at depth. The aim of this study was to report changes in the adenosinergic system induced by exposure to extreme oxygen partial pressures. At high altitudes, the increased adenosine concentration contributes to brain protection against hypoxia through various mechanisms such as stimulation of glycogenolysis for ATP production, reduction in neuronal energy requirements, enhancement in 2,3-bisphosphoglycerate production, and increase in cerebral blood flow secondary to vasodilation of cerebral arteries. In the context of mountain illness, the increased level of A_2A_R expression leads to glial dysfunction through neuroinflammation and is involved in the pathogenesis of neurological disorders. Nonetheless, a high level of adenosine concentration can protect against high-altitude pulmonary edema via a decrease in pulmonary arterial pressure. The adenosinergic system is also involved in the acclimatization phenomenon induced by prolonged exposure to altitude hypoxia. During hyperoxic exposure, decreased extracellular adenosine and low A_2A_ receptor expression contribute to vasoconstriction. The resulting decrease in cerebral blood flow is considered a preventive phenomenon against cerebral oxygen toxicity through the decrease in oxygen delivery to the brain. With regard to lung oxygen toxicity, hyperoxia leads to an increase in extracellular adenosine, which acts to preserve pulmonary barrier function. Changes in the adenosinergic system induced by exposure to extreme oxygen partial pressures frequently have a benefit in decreasing the risk of adverse effects.

## 1. Introduction

Healthy subjects experience major oxygenation changes during various recreational or professional activities performed under ambient pressure that is different than atmospheric pressure. Although the composition of environmental air is constant with approximately 78% nitrogen and 20.9% oxygen, climbers and aviators are acutely exposed to decreases in inspired partial pressure of oxygen (O_2_), secondary to the decrease in ambient pressure induced by elevated altitudes. Conversely, divers and professional workers in tunnel boring machine hyperbaric chambers are subject to an increase in ambient pressure and are exposed to hyperoxia. Changes in oxygenation conditions induce major alterations in cardiorespiratory function in resting healthy subjects. 

Adenosine is a nucleoside that mainly comes from the dephosphorylation of ATP and whose extracellular concentration depends on the energy state of the tissues and the degree of oxygenation or inflammation [1]. Thus, it is recognized that the oxygenation state has a strong impact on adenosine plasma levels (APLs). Hypoxia induces an increase in APLs [2,3], whereas hyperoxia leads to a decrease in APLs [4,5]. The hypoxia-induced adenosine increase is mainly due to the hypoxia-inducible factor (HIF) pathway because this transcription factor (where the alpha subunit is stabilized in hypoxic conditions) inhibits the biosynthesis of adenosine kinases, leading to the accumulation of adenosine in the intra- and extracellular spaces [6].

Because adenosine, through the activation of its G-coupled membrane receptors, named P1 receptors, strongly impacts heart rate and vasodilation, changes in APLs contribute to physiological adaptation and acclimatization under extreme oxygenation conditions. In addition, healthy subjects most often exercise during their professional or leisure activities. Because adenosine also plays a role in exercise-induced hemodynamic changes [7], APL-related interactions between exercise and oxygenation changes could occur. 

The aim of this short review was, therefore, to report on the contribution of adenosine to physiological alterations and pathological risks of healthy subjects submitted to extreme oxygenation conditions induced by changes in ambient pressure.

### 1.1. Altitude Hypoxia

The decrease in ambient pressure secondary to the ascent to altitude exposes the body to hypoxia. For example, the partial pressure of O_2_ decreases from 19.6 KPa at sea level to 6.5 KPa at the summit of Everest. Apart from mountaineers, a significant part of the population lives at an altitude of more than 3000 m with a partial pressure of O_2_ around 85 to 110 mmHg, with physiopathological consequences [8] involving the adenosinergic system. Acclimatization to severe hypoxia is achieved by an increase in O_2_ delivery or a decrease in O_2_ requirements to maintain the critical O_2_ tension at which cell function is not impaired. Healthy subjects such as climbers and aviators experience hypoxia at high altitudes. In the areas where compensation is possible, the immediate physiological response to hypoxia in healthy subjects includes increases in breathing rate, diuresis and cardiac output, and erythropoiesis when exposure to hypoxia continues for several weeks. Some cells are particularly sensitive to the decrease in partial pressure of O_2,_ such as neurons. Acute hypoxic exposure can lead to impaired cognitive function and sometimes loss of consciousness and seizures. To protect cerebral function, early increases in cerebral blood flow (CBF) occur [9]. The magnitude of the changes in CBF is related to the changes in cerebral vasomotion secondary to hypoxic and hypercapnic ventilatory responses. Hypoxia leads to vasodilation, while hypocapnia induces vasoconstriction. 

It was reported that brain adenosine blood concentration increases during hypoxia [10]. Furthermore, an increase in APLs is common during hypoxia, ischemia, inflammation, and beta-adrenergic stimulation [1,2,11,12,13], and was observed in healthy volunteers exposed to altitude hypoxia [14]. The mechanism supporting the increase in APLs in such circumstances is well-documented. An increase in the activity of soluble ecto-5′-nucleotidae (CD73), an enzyme that hydrolyzes AMP into adenosine, was reported. Thus, elevated CD73 contributes to hypoxia-induced adenosine accumulation [15].

Finally, other ectonucleotidases, including NTDPase 1(CD39), which hydrolyzes ATP into AMP, or NTPDase 2, which converts ATP into ADP, may participate in the adaptive reaction to hypoxia. Both are present in the cardiovascular system and modulate ligand concentration for P1 and P2 receptors [16]. 

A decrease in equilibrative nucleoside transporter 1 (ENT-1) expression was reported among apnea snorkelers. This decrease in ENT-1 contributes to the adenosine accumulation in extracellular spaces [17]. Adenosine acts on a number of tissues through the activation of four G-coupled membrane receptors, named A_1_R, A_2A_R, A_2B_R, and A_3_R, as a function of their pharmacological properties and primary sequence. Activation of A_1_R or A_3_R inhibits adenylate cyclase activity, leading to cAMP production inhibition, while A_2A_Rs and/or A_2B_Rs activation leads to the production of cAMP in target cells [18,19]. While A_1_ and A_2A_ exhibit high affinity for adenosine [18,19], A_2B_ is known as a low-affinity receptor [20]. Through A_2B_ activation and protein kinase A (PKA) phosphorylation, a rapid ubiquitination and proteasomal degradation of ENT-1 was observed during faster acclimatization to high altitude that led to the inhibition of adenosine uptake by erythrocytes and to the increase in extracellular adenosine concentration [14].

Previous studies have supported the contribution of increased extracellular adenosine concentration to protecting the brain from hypoxia [2,21]. This protection includes the stimulation of glycogenolysis for ATP production via anaerobic glycolysis and the reduction in neuronal energy requirements [22]. Furthermore, the increase in APLs and CD73 activity can increase via A_2B_ receptor activation, 2,3-bisphosphoglycerate (2,3-BPG) production, which decreases the affinity of hemoglobin for dioxygen, promoting O_2_ delivery to tissues [15]; see Figure 1. 

The increase in adenosine extracellular concentration is a major actor in hypoxia-induced vasodilation through activation of A_2A_ and A_2B_ receptors [23,24]. Vasodilation contributes to cerebral blood flow increase. Meanwhile, cardiac myocytes secrete adenosine, directly acting on the vascular smooth muscle, reducing vascular tone, and increasing blood flow in the coronary circulation [25]. Increased concentrations of interstitial adenosine in skeletal muscles was also observed in healthy volunteers submitted to acute systemic hypoxia [26], resulting in increased muscle blood flow. 

Lastly, A_1_R seems to have protective effects on brain function during intermittent hypoxia in mice. The activation of A_1_R helps to protect hippocampal neurons against apoptosis, enhances long-term potentiation, and finally enhances learning and memory. These effects occur via upward regulation of a C-protein kinase (PKC) [27].

### 1.2. Acute Altitude Illness

Acute altitude illness consists of clinical disorders that occur during the first days (from some hours to 5 days) after ascent to altitudes above 2500 m in unacclimatized subjects. Various clinical forms of injury, such as acute mountain sickness, high-altitude cerebral edema, and high-altitude pulmonary edema, are observed [28]. 

The main factor responsible for mountain illness is the decrease in arterial partial pressure of O_2_ at high altitudes. An increased risk of injury was demonstrated in subjects with a poor increase in the ventilatory response to hypoxia [29], resulting in a major decrease in partial pressure of O_2_ in arterial blood. Furthermore, physical activity increases the risk of a bad tolerance to altitude [30]. The acute symptoms of mountain sickness include various disorders such as headache, fatigue, loss of appetite, nausea or vomiting, sleep disorders, dizziness, and confusion. Progression from acute mountain sickness to high-altitude cerebral edema (HACE) can occur, especially if clinical disorders are neglected, and the subject continues to ascend. HACE is characterized by various degrees of neurological disorders such as confusion, ataxia, psychiatric disorders, and disturbances of consciousness that can progress to a deep coma. 

The severity of neurological disorders at high altitudes can be extremely different depending on the individual. Nevertheless, cognitive impairment is common and correlated with altitude [31]. Neuroinflammation, including glial cell activation and cytokine release in the central nervous system, is the main factor leading to cognitive impairment in hypobaric hypoxia [32,33]. Adenosine, through the activation of its receptors, was shown to participate in altitude-induced disorders of the central nervous system. Acute hypobaric hypoxia is associated with an increase in A_2A_R expression in hippocampal microglia [33]. It is recognized that A_2A_R activation leads to glial dysfunction through neuroinflammation [34] and glutamate-mediated excitotoxicity [35].

The adenosinergic system also has an impact on the lung response to hypoxia. An upregulation of A_1_R was observed during experimental hypoxia in rats [36]. Lungs are also threatened during altitude hypoxia in humans. High-altitude pulmonary edema (HAPE) is a life-threatening disorder that can occur in healthy unacclimatized individuals. HAPE is a noncardiogenic pulmonary edema. The key pathogenic mechanism is exaggerated pulmonary hypertension induced by hypoxia. Contributing factors such as inflammation, endothelial dysfunction, sympathetic overactivity, and fluid retention were cited [37]. Interestingly, young and trained subjects do not have a lower risk. Adenosine may provide protection against HAPE. In an experimental study performed on animals submitted to hypoxia, Mentzer et al. reported that adenosine infusions entirely abolished the increase in pulmonary vascular resistance [38]. Because a large individual susceptibility was reported, further studies should be of interest in assessing the contribution of the adenosinergic system in protection against HAPE. Finally, it seems that adenosine does not influence the energy substrate utilization during exercise in hypoxia in humans. It was reported that glucose metabolism and uptake increased under hypoxia conditions (corresponding to 3400 m of altitude) during exercise. Nevertheless, the increase in glucose uptake did not appear to be modulated by adenosine through A_1_ or A_2_ receptor subtypes. The glucose uptake in the human quadriceps femoris was not altered by theophylline, a nonselective antagonist of adenosine receptors [39].

### 1.3. Chronic Hypoxia: Life at Altitude

Prolonged exposure to altitude hypoxia is known to result in the development of an acclimatization phenomenon. In a human study, Song et al. [14] reported that the adenosinergic-signaling network enhanced the hypoxia adenosine response to counteract hypoxia-induced maladaptation. They measured APLs and soluble CD73 activity in healthy subjects at sea level and during a stay at high altitudes (5260 m). The volunteers returned for several days (from 7–21 days) at 1525 m, and a further blood sample was collected upon re-ascent at 5260 m. The authors found that APLs and CD73 activity were significantly higher upon re-ascent to 5260 m for 1 day, after spending several days at 1525 m, compared with the first hypoxia exposure. Consequently, the first stay at high altitude can enhance the defense response to hypoxia through an increase in APLs and CD73 activity.

Although adenosine production is activated during prolonged stays at high altitudes, its action on the artery decreases. Calbet et al. [40] reported that short-term residence at altitude (between 8 and 12 days at 4554 m) induced an increase in resting blood pressure. Vasodilatory responses secondary to exogenous adenosine infusion were impaired by alteration in endothelial function. Thus, chronic or acute exposures to a high endogenous adenosine extracellular concentration may have different effects on the cardiovascular system.

## 2. Hyperoxia

In hyperbaric conditions, subjects breathe a mixture of high-pressure gases through a regulator. Most frequently, diving tanks contain compressed air (filtered and dehumidified), i.e., 78% nitrogen, 20.9% oxygen, and small proportions of trace gases. The increase in ambient pressure generates an increase in the partial pressures of oxygen and nitrogen. Furthermore, the gas density is increased. Some divers inhale oxygen-enriched gas mixtures to decrease the nitrogen content of tissues and blood at the time of decompression and to limit the occurrence of decompression sickness. For an air dive, the partial pressure of O_2_ is 40 KPa at 10 m depth and 60 KPa at 20 m depth. In the particular case of military diving, the use of pure oxygen through a closed-circuit self-contained underwater breathing apparatus (SCUBA) allows better self-sufficiency (the exhaled gas is reused after the CO_2_ is extracted by lime) and discretion (no bubbles). Consequently, hyperoxia is a constant stressor for healthy subjects working in hyperbaria, such as SCUBA divers or professional workers in a tunnel boring machine hyperbaric chamber.

### 2.1. Cardio-Vascular Changes

Hyperoxic exposure has a major impact on cardiovascular function in healthy subjects. Numerous studies have shown that cardiovascular responses to acute hyperoxia include a decrease in cardiac output related to the simultaneous decreases in heart rate and stroke volume [41,42,43]. Increases in mean blood pressure and systemic vascular resistance, and a decrease in arterial compliance, have been documented in resting healthy volunteers breathing pure oxygen [41,44,45]. Such an effect of oxygen appears to be related to its vasoconstrictive action on the peripheral vascular system. The exact mechanism by which hyperoxia induces vasoconstriction is not fully understood. The increase in partial pressure of O_2_ and the production of reactive oxygen species can contribute to arterial vasoconstriction through an alteration in endothelial function or a direct effect on the vascular smooth muscle [46,47]. However, the adenosinergic system is likely involved in oxygen-induced vasoconstriction. An increase in systolic blood pressure (SBP) associated with a decrease in APL was observed in healthy volunteers breathing pure oxygen compared with medical air [5]. The lowering of APLs and the increase in SBP were also reported during orthostatic stress under hyperoxia. In this experiment, two volunteers fainted during orthostatic stress while breathing medical air. An increase in APL was found during the test. In contrast, during hyperoxic exposure, they suffered from no clinical troubles, and APLs were low compared with the medical air condition. The vasoconstrictor effect of the decrease in APL was therefore implicated in hemodynamic differences and better tolerance to orthostatic stress during hyperoxic exposure [5]. These findings were supported by another study in healthy volunteers reporting that breathing pure oxygen at atmospheric pressure can increase peripheral vascular resistance associated with a decrease in APLs compared with the same session in air [48]. The differences remained significant between the two sessions when a low-intensity exercise was added to each experimental condition. Meanwhile, hyperoxia did not suppress the increase in APL and the decrease in peripheral vascular resistance commonly recorded during exercise compared with resting reference [48]. In addition to various factors such as nitric oxide, prostacyclin (PGI2), and endothelium-derived hyperpolarizing factor [49], adenosine may be another important contributor involved in muscle hyperemia during exercise [50].

### 2.2. Oxygen Toxicity 

Hyperoxia is known to expose healthy volunteers to various risks. Short acute exposure to the high partial pressure of O_2_ can induce cerebral toxicity leading to seizures. Additionally, prolonged exposures can damage lung function. 

#### 2.2.1. Nervous System Toxicity

Retinal toxicity is one of the first indicators of hyperoxia [51]. From this perspective, the lack of A_1_R reduced hyperoxia-induced retinal toxicity in mice [52], suggesting that A_1_R activation did not protect against hyperoxia-induced retinal toxicity. 

Brain oxygen toxicity is linked to oxidative stress induced by hyperoxia. Damage is secondary to increased production of reactive oxygen species (ROS) and/or reactive nitrogen species and lipid peroxidation, which impair cell membranes [53]. Clinical disorders include disturbances of vision (tunnel vision), headache, nausea, muscle twitching, and convulsions similar to epileptic seizures with loss of consciousness [54]. 

From the blood samples of animals submitted to normobaric or hyperbaric hyperoxia, Bruzzese et al. reported a decrease in extracellular adenosine attributed to the degradation into inosine by serum amino-hydrolase ADA [4]. They also monitored the effects of hyperoxia on A_2A_R expression in the brain, using mRNA expression of A_2A_R, and found that hyperoxic exposure resulted in low A_2A_R mRNA expression and low A_2A_R production levels. It was previously reported that the lack of A_2A_R activation resulted in vasoconstriction [55]. Conversely, activation of A_2A_R leads to vasodilation via stimulation of potassium channels, mainly K_V_ and K_ATP_ channels [18,19].

Nitric oxide can contribute to central nervous system oxygen toxicity via a vasodilating action, leading to an increase in oxygen supply to tissue and its subsequent deleterious consequences through an increase in ROS production [56]. In contrast, changes in the adenosinergic system recorded during hyperoxic exposure may be an adaptive response because vasoconstriction may be preventive against cerebral oxygen toxicity through attenuating oxygen delivery [57,58].

#### 2.2.2. Lung Toxicity

It has long been known that prolonged exposure to hyperoxia can lead to lung damage (Lorrain-Smith effect) [59]. Pulmonary disorders consist of three successive phases, i.e., inflammatory, proliferative-reparative, and fibrotic. The severity is positively correlated with the duration of exposure and the level of partial pressure of O_2_. Hyperoxia can disrupt the structure and function of the pulmonary epithelial barrier through the destruction of the pulmonary epithelial tight junction structures (see Figure 2). This impairment was attributed to the downregulation of tight junction proteins such as occludin, zonula occludens-1, and claudin-4 [60]. Furthermore, Xu et al. [61] found that the downregulation of tight junction proteins was mediated by caveolin-1. This protein is a component of plasma membranes and is found in many cells such as epithelial cells, endothelial cells, smooth muscles, and alveolar macrophages. Caveolin 1 plays a major role in regulating lung cell functions [62].

An increase in the extracellular concentration of adenosine was observed after acute lung injury [63,64]. This finding was later confirmed in animals submitted to hyperoxia [61]. A high extracellular adenosine concentration may be protective through decreased inflammatory response and improved endothelial barrier function [65]. The action of adenosine (Figure 2) appears partially mediated by the activation of adenosine A_2B_R, which protects occludin levels in the lungs [66].

The increase in extracellular adenosine concentration, secondary to CD73 activation, counteracts inflammation (through the production of ROS) and protects barrier function by activating adenosine A_2B_ receptors.

The impact of adenosine and A_2A_R on the protection against lung oxygen toxicity is also supported by human studies. The high inspired fraction of oxygen used to treat patients suffering from acute respiratory distress syndrome could weaken the endogenous anti-inflammatory mechanism by decreasing the A_2A_R signaling pathway [67]. Based on the benefit of intratracheal injection of CGS21680 in mice, some authors suggested using inhalation of A_2A_R agonists during oxygen therapy to reduce the risk of lung toxicity [67].

## 3. Conclusions

Adenosine concentration and adenosine receptor activity are altered by changes in ambient oxygen pressure experienced by healthy subjects during professional or leisurely activities such as climbing or diving. When climbing at a high altitude, the increase in adenosine concentration can contribute to improved tolerance to hypoxia. In contrast, the increase in the expression of A_2A_R can promote neurological disorders involved in mountain illness. During hyperoxic exposure, changes in the adenosinergic system lead to vasoconstriction and may decrease the risk of cerebral oxygen toxicity. The adenosinergic system is also recognized as a protector against the lung toxicity of oxygen. Finally, acute and chronic exposures to a high endogenous adenosine extracellular concentration lead to different adaptation mechanisms. Because the involvement of the adenosinergic system seems decisive in some diseases induced by major changes in the partial pressure of O_2_, the use of drugs that modulate this system may be of interest to treating or prevent clinical disorders.

## Figures and Tables

**Figure 1 biomedicines-10-02059-f001:**
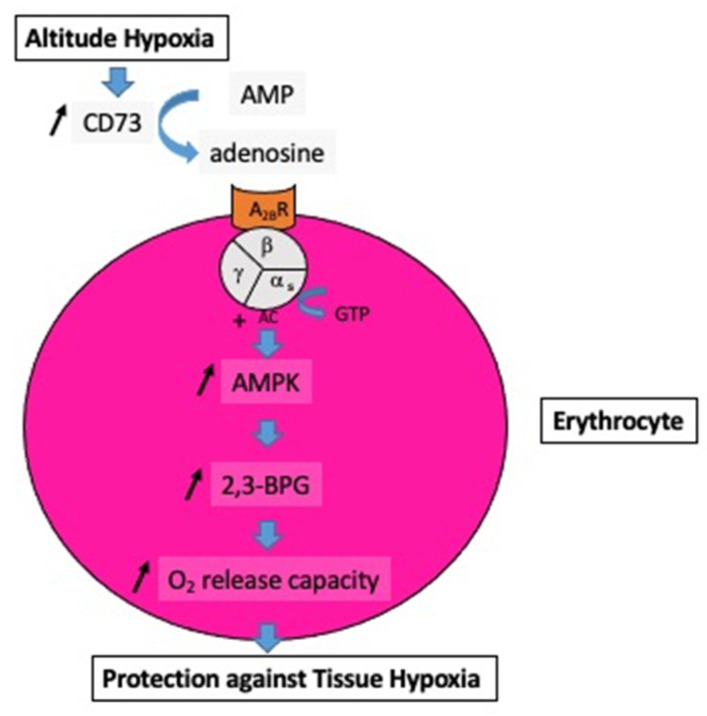
Impact of the increase in extracellular adenosine on the oxygen release capacity from erythrocytes. At high altitudes, hypoxia increases the activity of soluble ecto-5′-nucleotidase (CD73), which converts AMP into adenosine. The subsequent increase in extracellular adenosine induces, via the erythrocyte A_2B_ receptor (A_2B_R) and via the complex guanosine triphosphate (GTP-alpha s subunit of the heterotrimeric G protein), an increase in adenyl cyclase activity (AC) followed by activation of AMP-activated protein kinase (AMPK), and finally the production of 2,3-bisphosphoglycerate (2,3-BPG), which enhances the oxygen (O_2_) release capacity to peripheral tissues.

**Figure 2 biomedicines-10-02059-f002:**
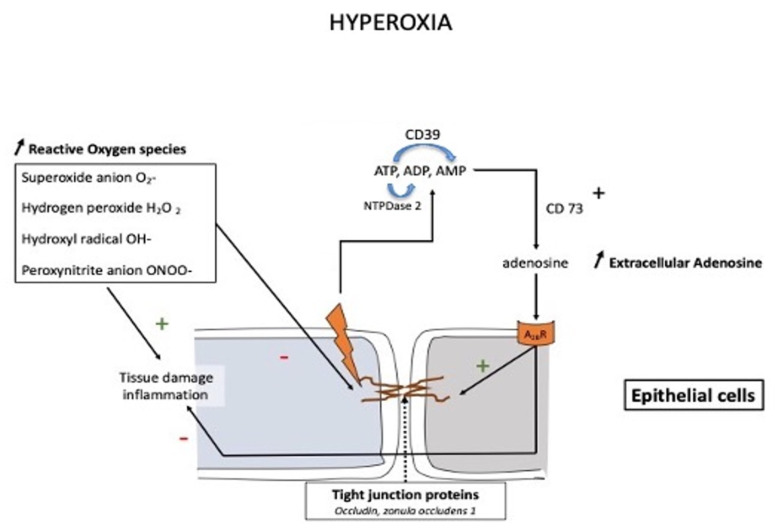
Schematic presentation of the protective action of the adenosine system in pulmonary oxygen toxicity. Prolonged exposure to high oxygen partial pressure leads to an impairment in pulmonary barrier function through inflammation and disruption of the tight junction via a downregulation of tight junction proteins such as occludin, zonula occludens-1, and claudin-4. The increase in adenosine is mainly due to the activation of CD73, which converts AMP into adenosine, while other nucleotidases such as CD39 (NTDPase1) or NTDPase2 may also participate in the modulation of extracellular adenosine levels.

## Data Availability

Not applicable.

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
