# Peer review of "Contribution of Adenosine in the Physiological Changes and Injuries Secondary to Exposure to Extreme Oxygen Pressure in Healthy Subjects"

_biomedicines, 2022, doi:10.3390/biomedicines10092059_

Round 1

Reviewer 1 Report

General comments

Boussuges and colleagues present a review entitled “Contribution of Adenosine in the physiological changes and injuries secondary to exposure to extreme oxygen pressure in healthy subjects”.

The authors reviewed the recent literature about the changes of the adenosinergici system during hypoxia and hyperoxia conditions from different points of view (lung, brain and cardio-vascular system).

The manuscript is well written and organized. In my opinion, this article responds to the scope of Biomedicines and should be of interest and useful for its readership. Anyway, the manuscript requires minor revision, to be clearer and more exhaustive for the reader.

1.     Please, add a citation in 2.1 paragraph (Altitude hypoxia).

2.     Please, add some citation and phrases about the low affinity of A2BR for adenosine and adenosine metabolism in introduction.

3.     Specify the acronym of “ENT-1” the first time that it is mentioned (paragraph 2.2 – Physiological adaption).

4.     The following phrase at pag.2 is better to move after you are mentioned all adenosine receptors: “Furthermore, through A2B activation and PKA phosphorylation, a rapid ubiquitination and proteasomal degradation of ENT-1 was observed during faster acclimatization to high altitude that leads to the inhibition of adeno-sine uptake by erythrocytes, and to the increase in extracellular adenosine concentration [13].”

5.     In the last phrase of the paragraph 2.3 (Acute altitude illness) specify where glucose uptake was measured. The phrase is: “Indeed, while glucose metabolism and uptake increase in hypoxia condition (corresponding to 3400 m of altitude) during exercise, the increase in glucose uptake was not modulated by adenosine since theophylline a non-selective adenosine receptor antag-onist did not modify the glucose uptake [36].”

6.     Page 4, abbreviate “plasma adenosine levels” with APL. The phrase is: “They measured plasma adenosine levels and soluble CD73 in normal individuals at sea level and during a stay at high altitude (5260 m).”

7.     Please, check the abbreviation of adenosine receptors, are not always the same.

8.     Check the references list.

Author Response

Thank you for giving us the opportunity to improve the mansucript.

The work has been modified in accordance

Yours Sincerely

R Guieu

Reviewer 2 Report

The review publication submitted for peer-review focuses on the influence of adenosinergic system on the (patho)physiological state under extreme conditions of healthy human habitation: from altitude hypoxia to underwater hyperoxia, based on 64 literature references. The review is well divided / composed for particular paragraphs, describing if not exhaustively, then in a clear way the influence on adenosinergic system in particular systems. The overall feeling of the manuscript is good; however I found some minor inconsistences, and/or editorial and grammatical errors.

Page 2, top

“..due to the HIF pathway..” please explain the abbreviation. A list of abbreviations (eg. at the end of the Manuscript) used in this review would be of great use.

Page 2, middle

“2.1 Altitude Hypoxia” and “2.2. Physiological adaptation” - there is no need for starting of new paragraphs with the second sequences.

Page 2, bottom

ENT-1 - please elaborate on the abbreviation

Page 3, top

“..leading to cAMP production inhibition while A 2A/B Rs..” – using of “A2A and/or A2B receptors” would be of higher clarity.

Figure 1 description

(GTP) -alpha s – please use correct notation GTP-a-S or GTP-alpha-S

Page 3, bottom

Both paragraphs start from “lastly” – please change

PKC – please evaluate on the abbreviation

Page4, bottom

-       “the increase in glucose uptake was not modulated by adenosine since theophylline a non-selective adenosine receptor antagonist did not modify the glucose uptake [36].” – please clarify, since the inhibition of the activity of different adenosine receptors subtypes lead to different behavior, as stated earlier.

-       “They measured plasma adenosine levels and soluble CD73 in normal individuals” – what is “normal” individuals?

Page 5, mid

“…or 450 mmHg at 20m depth and even more so with mixture dives oxygenated type Nitrox” – please clarify

Page 6, bottom

-       “Xu et al [58] found that the downregulation of tight junction proteins was mediated by Caveolin-1.” Please elaborate on the Caveolin-1.

-       “[642he action of adenosine (Figure 2)”-  please correct

-        

Page 7, mid

“Some authors have suggested to use inhalation of A2A R agonists during oxygen therapy to decrease the risk of lung toxicity [64].” – please name a few of these inhalated agonists.

Last but not least, parts of the manuscript lack the a’s, an's and the’s, making it sometimes hard to go through the paragraphs and get the proper meaning. In my opinion the article should be proof-read by the native speaker or run through any grammar corrections software. The spacing (before/after hyphen, double spaces) and typos should be checked as well.

Author Response

Thank You for giving us the opportunity to improve the mansucript.

The work has been modified in accordance

Sincerely Yours

R Guieu

Reviewer 3 Report

Boussuges A  et al. at MS titled Contribution of Adenosine in the physiological changes and injuries secondary to exposure to extreme oxygen pressure in healthy subjectsreport the contribution of adenosine to the physiological alterations and pathological risks of healthy subjects submitted to ex-treme oxygenation conditions, induced by changes in ambient pressure. By sure it is important work from the physiological point of view but it cannot be accepted without biochemical major revision. Authors should increase their enzymological knowledge. CD73 hydrolyse AMP to adenosine not ATP to AMP! Cited (number 14) work is clearly describing AMP as 5`NT substrate. The conclusions should be reformulate/rethink according to correct enzyme mechanism. As authors start their analyses from ATP tey should include also other ectonucleotidases present in cardiovascular.

I have also a few minor comments:

1.       Page 2, please change “soluble ecto-5’-nucleotidae” to “soluble ecto-5’-nucleotidase”.

2.      Please verify all typing errors (lack of space etc.).

3.      The nomenclature of purinoceptors for P1 should be verified.

4.      Fig. 2 should include also other than 5’NT ectonucleotidases to justify hydrolyses of ATP and ADP. 

Author Response

Thank You for your comments and suggestions

The work has been modified in accordance

Hoping this new version meets your approval

R Guieu

Round 2

Reviewer 3 Report

Authors corrected the MS accordingly to the Reviewers’ comments. MS can be accepted for publication in actual form.